# Modulation of the Mucosa-Associated Microbiome Linked to the PTPN2 Risk Gene in Patients with Primary Sclerosing Cholangitis and Ulcerative Colitis

**DOI:** 10.3390/microorganisms9081752

**Published:** 2021-08-17

**Authors:** Luisa Denoth, Pascal Juillerat, Andreas E. Kremer, Gerhard Rogler, Michael Scharl, Bahtiyar Yilmaz, Sena Bluemel

**Affiliations:** 1Department of Gastroenterology and Hepatology, University Hospital Zurich, Rämistrasse 100, 8091 Zürich, Switzerland; luisa.denoth@hotmail.ch (L.D.); Andreas.Kremer@usz.ch (A.E.K.); Gerhard.Rogler@usz.ch (G.R.); Michael.Scharl@usz.ch (M.S.); 2Maurice Müller Laboratories, Department for Biomedical Research, University of Bern, 3012 Bern, Switzerland; pascal.juillerat@insel.ch (P.J.); Bahtiyar.Yilmaz@dbmr.unibe.ch (B.Y.); 3Department of Visceral Surgery and Medicine, Bern University Hospital, University of Bern, 3012 Bern, Switzerland; 4Department of Medicine 1, Friedrich-Alexander-University Erlangen-Nürnberg and University Hospital Erlangen, 91054 Erlangen, Germany

**Keywords:** PSC, PTPN2, TCPTP, mucosa-associated microbiome, *Roseburia*, *Tepidimonas*, *Actinobacillus*, *Haemophilus*, *Fusobacterium*, *Brachyspira*, *Eubacterium*

## Abstract

Gut microbiota appears to be involved in the pathogenesis of primary sclerosing cholangitis (PSC). The protein tyrosine phosphatase nonreceptor 2 (PTPN2) gene risk variant rs1893217 is associated with gut dysbiosis in inflammatory bowel disease (IBD), and PTPN2 was mentioned as a possible risk gene for PSC. This study assessed the microbial profile of ulcerative colitis (UC) patients with PSC and without PSC (non-PSC). Additionally, effects of the PTPN2 risk variant were assessed. In total, 216 mucosal samples from ileum, colon, and rectum were collected from 7 PSC and 42 non-PSC patients, as well as 28 control subjects (non-IBD). The microbial composition was derived from 16S rRNA sequencing data. Overall, bacterial richness was highest in PSC patients, who also had a higher relative abundance of the genus *Roseburia* compared to non-PSC, as well as *Haemophilus*, *Fusobacterium*, *Bifidobacterium,* and *Actinobacillus* compared to non-IBD, as well as a lower relative abundance of *Bacteroides* compared to non-PSC and non-IBD, respectively. After exclusion of patients with the PTPN2 risk variant, *Brachyspira* was higher in PSC compared to non-PSC, while, solely in colon samples, *Eubacterium* and *Tepidimonas* were higher in PSC vs. non-IBD. In conclusion, this study underlines the presence of gut mucosa-associated microbiome changes in PSC patients and rather weakens the role of PTPN2 as a PSC risk gene.

## 1. Introduction

Primary sclerosing cholangitis (PSC) is a rare cholestatic liver disease causing intra- and extrahepatic bile duct strictures and fibrosis, which ultimately leads to liver cirrhosis [1,2]. PSC is often associated with ulcerative colitis (UC), affecting roughly 4% of all UC patients [3,4,5]. While the exact pathogenesis of PSC remains largely unknown, UC is the main risk factor, with 60–80% of PSC patients suffering from concurrent UC [4,6,7,8].

In recent years, changes in the microbiome, metabolome, and intestinal barrier function were extensively studied to unravel the pathogenesis of PSC [9]. This system is often referred to as “gut–liver axis”, which symbolizes the interaction of gut microbiota and the liver: Intestinal dysbiosis, i.e., the change in the microbiota linked to disease, may weaken the intestinal barrier and lead to translocation of harmful bacterial particles, which in the end leads to liver inflammation and fibrosis. It is thought that intestinal dysbiosis in PSC patients with UC leads to translocation of pathogenic microbes or their products, which not only cause liver inflammation, but also biliary strictures [10,11,12,13]. Studies investigating changes in the fecal microbiota showed a distinctly different bacterial composition in PSC patients compared to healthy controls and UC patients [14,15,16], with certain bacterial taxa such as *Fusobacterium* associated with intestinal inflammation, and taxa such as *Enterococcus* rather associated with cholangitis [17]. This suggests a strong link between the composition of the gut microbiome and the pathogenesis of UC and PSC, respectively. Only few studies with inconsistent findings investigated the mucosa-associated microbiome in PSC [18]. Furthermore, changes in the gut microbiota profile resulting in altered bile acid homeostasis and increased intestinal inflammation contribute to the pathogenesis of PSC [19,20].

A dysfunction of protein tyrosine phosphatase nonreceptor 2 (PTPN2), caused by the single nucleotide polymorphism (SNP) rs1893217, is associated with intestinal dysbiosis and a more severe disease course in inflammatory bowel disease (IBD) patients [21,22,23]. While the homozygous mutation of PTPN2 occurs in roughly 1.5% of the normal population, it is present in 3.8% of IBD patients [24]. PTPN2 has also been discussed as a risk gene for PSC [25]. However, it remains elusive whether this SNP is also associated with intestinal dysbiosis in PSC patients as reported in IBD patients.

We hypothesized that the presence of PTPN2 SNP rs1893217 is associated with intestinal dysbiosis in PSC patients. To this aim, the study compared the mucosa-associated microbiota of UC patients with and without PSC according to the presence of the PTPN2 SNP, in reference to a control group, by analyzing mucosal biopsies from multiple intestinal locations from UC patients of the Swiss IBD Cohort (SIBDCS) and a (local) Bern cohort.

## 2. Materials and Methods

Sample dataset: This study analyzed biopsies from the terminal ileum, the right colon, the left colon, and the rectum collected during ileocolonoscopies, as published previously [26]. The SIBDCS comprises primarily IBD patients, who may simultaneously suffer from PSC, which is a rare liver disease. Therefore, data from seven UC patients with PSC (PSC) could be retrieved from the SIBDCS. A total of 42 UC patients without PSC (non-PSC) and 28 controls (non-IBD) were selected based on matching age, gender, and Modified Truelove and Witts activity index (MTWAI) values of the PSC patients, if applicable [26]. The asymptomatic control subjects underwent a screening ileocolonoscopy for colorectal carcinoma, were negative for all other biochemical and hematological tests, and presented without any macroscopic or microscopic abnormalities [26].

Ethics statement: This study is a sub-study of the Swiss IBD Cohort Study. Patient biopsy samples and clinical data collection from patients of the SIBDCS were approved by the Ethics Committee of the Canton Zürich (KEK-ZH-Nr. 2013-0284). Control data were retrieved from the Bern Human Intestinal Community project, approved by the Ethics Commission of the Canton Bern (KEK-BE-Nr 251/14 and 336/14). Informed consent for data collection and analysis was obtained from all patients of both study cohorts.

DNA extraction: Biopsies were collected in 2 mL microcentrifuge tubes (Sigma-Aldrich, St. Louis, MO, USA) and stored at −80 °C until DNA extraction, as published previously [21]. Total DNA was isolated using AllPrep DNA/RNA Mini Kit (Qiagen, Hilden, Germany) according to the manufacturer’s instructions. In brief, 600 μL of RLT lysis buffer (Qiagen), plus β-mercaptoethanol and a metal bead were added into each tube. Samples were then homogenized using the Retsch Tissue Lyser (Qiagen) at 30/s for 3 min and 3 min centrifugation at 13,000× *g*. Total supernatants were transferred into a Prep DNA mini spin column (Eppendorf, Hamburg, Germany) and centrifuged at 9000× *g* for 30 s. DNA attached to spin columns was washed and desalted using 500 μL of Buffer AW1 and Buffer AW2 (Qiagen). Total DNA was eluted in 30 μL RNase-free water into 1.5 mL microfuge tubes. The concentration and purity of the isolated DNA was analyzed using NanoDrop^®^ (Thermo Fisher Scientific, Waltham, MA, USA).

16S rRNA sequencing: Amounts of 500 to 1500 ng of DNA per sample were used to amplify the V5/V6 region of the 16S rRNA gene. The expected product length was ~350 bp including adaptors and barcodes. Bacteria-specific primers (forward 5′ CCATCTCATCCCTGCGTGTCTCCGACTCAGC barcode ATTAGATACCCYGGTAGTCC 3′ and reverse 5′ CCTCTCTATGGGCAGTCGGTGATACGAGCTGACGACARCCATG-3′) were used [27]. PCR conditions consisted of an initial 5 min at 94 °C denaturation step, followed by 35 cycles of 1 min denaturation at 94 °C, 20 s annealing cycle at 46 °C, and 30 s extension cycle at 72 °C, with a final extension for 7 min at 72 °C. Samples were kept at 4 °C until loading onto a 1% agarose gel. Amplicons were then purified using the Gel Extraction Kit (Qiagen), and the pooled amplicon library at 26 pM concentration was used for sequencing, which was performed within the Ion PGMTM System (Thermo Fisher) using an Ion PGMTM Sequencing 400 Kit and an Ion 316TM Chip V2 [28].

Computation analysis of the 16S rRNA microbial data: Combined FASTQ sequencing files were first processed in QIIME 1.9.1 pipeline, as described [29] using custom analysis scripts for analysis on the UBELIX Linux cluster of the University of Bern [26]. Samples with more than 5000 high-quality reads were then used for downstream analysis in QIIME and R (R Foundation for Statistical Computing, Vienna, Austria). Operational taxonomic units were picked using UCLUST with a 97% sequence identity threshold and followed by taxonomy assignment using the latest Greengenes database (version gg_13_5; greengenes.secondgenome.com (accessed on 1 February 2021)). The operational taxonomic unit (OTU) abundance biome file and mapping file were used for statistical analyses and data was visualized with the phyloseq R package [30]. Species richness within samples was calculated using the α-diversity indices Simpson and Shannon. β-diversity between samples was calculated using Bray–Curtis genus-level community dissimilarities. Mann–Whitney U tests were performed for α diversity and Adonis (PERMANOVA) for β diversity was performed as a statistical test to confirm the strength and statistical significance of groups in the same distance metrics in the QIIME pipeline and phyloseq in R [30,31]. Taxonomy profile of samples was performed using multivariate analysis by linear models R package [32] to find associations between tested groups. The *q*-value package was implemented in MaAsLin2 to correct for multiple testing (Benjamini–Hochberg false discovery rate correction; a false discovery rate (FDR), *q*-value of 0.2). Taxa present in at least 30% of the samples and that had more than 0.0001% of total abundance were set as cut-off values for further analysis. After correction for a false discovery rate, *q* < 0.05 was considered significant. Plots were generated with ggplot2 using phyloseq object.

## 3. Results

A total of 216 biopsy samples taken from the terminal ileum, right colon, left colon, as well as rectum, were analyzed by 16S rRNA sequencing. All included subjects were male. Table 1 summarizes the demographics of the included patients.

### 3.1. Differences in Bacterial Diversity

To obtain an overall impression of the bacterial diversity between groups, samples of the different locations were pooled and analyzed per patient. Results are given in Figure 1. α-diversity was higher in PSC patients compared to non-PSC patients and the control group (Figure 1A). The bacterial community in non-PSC patients and controls clustered broader than the microbiome of PSC patients (*p* < 0.001) (Figure 1B).

Firmicutes and Bacteroides are the most abundant bacterial phyla within the gut, and the ratio of these phyla is a sign of health status. Therefore, the Firmicutes/Bacteroidetes (F/B) ratio can be used as a measure for general microbiome changes based on the patients’ health status [33]. The F/B ratio was significantly higher in non-PSC and PSC compared to non-IBD (*p* < 0.001) (Figure 1C). Specific changes on the genus level are given below.

### 3.2. Gut Microbial Signature in UC Patients with PSC

Significant differences at the genus level for the pooled sampling sites (Figure 2) were observed for *Roseburia*, *Haemophilus*, *Fusobacterium*, *Bifidobacterium,* and *Actinobacillus,* and were higher in PSC patients compared to non-IBD, while the genus *Bacteroides* was significantly lower in PSC patients. However, when analyzed for individual sampling site, no significant differences at the genus level were detected for PSC vs. non-IBD at the ileum, right colon, left colon, or the rectum.

### 3.3. Gut Microbial Signature in UC Patients without PSC

To distinguish which microbiome differences are mainly connected to UC, UC patients without PSC (non-PSC) were compared to controls (non-IBD). Significant differences for the pooled sampling sites are shown in Figure 3, with higher relative abundance of the genus *Dialister*, *Faecalibacterium*, *Blautia*, *Ruminococcus* (family: *Ruminococcaceae*) and *Roseburia* (all belonging to the phylum Firmicutes), and lower relative abundance of *Bilophila*, *Butyricimonas* and *Ruminococcus* (family: *Lachnospiraceae*) in non-PSC compared to non-IBD. Analyzed for individual sampling sites, *Ruminococcus* (family: Ruminococcaceae), *Blautia* and *Dialister* were higher at the left colon and *Roseburia* was higher at the ileum in non-PSC compared to non-IBD, while no significant differences at the genus level were detected at the right colon and rectum.

### 3.4. PSC Status in UC Results in Minor Taxonomic Changes

All investigated PSC patients suffered from concurrent UC. Hence, to detect microbiome changes associated solely with PSC, the PSC population was directly compared to UC patients without PSC (non-PSC).

When pooling the sampling sites, relative abundance of the genus *Bacteroides* was lower, while the abundance of *Roseburia* was higher in PSC compared to non-PSC (Figure 4). According to sampling sites, no significant differences at the genus level were detected for PSC vs. non-PSC at the ileum, right colon, left colon, or rectum.

### 3.5. Results According to Genetic Variation of PTPN2

PTPN2 has been discussed as risk gene for PSC [25]. UC patients carrying the C-allele of the PTPN2 SNP rs1893217 are more likely to have a more severe disease course, including gallstones—which suggests altered bile acid homeostasis [34]. As the C-allele is rather rare and was detected in only seven of the investigated patients (see Table 1), microbial differences between those patients and the wild-type carriers were not detected. However, after exclusion of the patients carrying the C-allele, the genus *Brachyspira* was significantly higher in PSC compared to non-PSC for the pooled sampling sites (*p* < 0.05; *q* < 0.05).

When analyzing the data in patients without the PTPN2 C-allele according to the origin of the biopsy, in the right colon the genus *Eubacterium* and *Tepidimonas* were increased in PSC compared to non-IBD, and *Dialister* was increased in non-PSC compared to non-IBD patients. Comparing PSC and non-PSC patients without the C-allele, no differences at the genus level were detected at the ileum, right colon, left colon, or the rectum.

## 4. Discussion

This study analyzed the mucosa-associated microbiome of 7 UC patients with PSC in comparison with 42 UC patients without PSC and 28 controls. PSC patients had an increased bacterial richness and a shift in the overall microbial composition. This became apparent in several differences at the genus level detected between groups. By exclusion of patients with a potentially disease-aggravating genetic variation of the PTPN2 gene, a significant increase of *Brachyspira* was detected in biopsies from PSC patients.

PSC patients had a higher Firmicutes/Bacteroidetes (F/B) ratio than UC patients without PSC and healthy controls (Figure 1). The F/B ratio describes the microbiome on the phyla level [35]. As those two phyla represent 90% of the gut microbiota, changes are considered a sign of dysbiosis [36]. An increase in the F/B ratio was associated with obesity [37]. In IBD and PSC patients, the F/B ratio was reduced in earlier studies [38,39]. The higher ratio in PSC patients might be related to the increase in *Roseburia*, a genus belonging to the phylum Firmicutes, detected in the presented study. In addition, earlier studies found very low rates of *Bacteroides* [40], which is in line with our findings, and might add to the increase in the F/B ratio. *Bacteroides* are producers of sphingolipids, which are crucial for intestinal homeostasis. A decrease in *Bacteroides* and therefore sphingolipid levels resulted in intestinal inflammation [41]. Another genus involved in intestinal pathologies is *Fusobacterium*. In our study, *Fusobacterium* was significantly higher in PSC compared to non-IBD. Previous studies support this finding as *Fusobacterium* was overrepresented in stool samples of PSC patients [17,42]. Furthermore, it is thought that *Fusobacterium* leads to a proinflammatory microenvironment and is ultimately associated with colorectal cancer [43]. Patients with UC have a high risk of developing these tumors, and this risk is increased by 30% in patients simultaneously suffering from PSC [44]. Together with the reduction of *Bacteroides* (and presumably sphingolipids), the elevation of *Fusobacterium* might contribute to this increased cancer risk.

Gut bacteria are involved in bile acid homeostasis [45,46,47]. Their enzyme bile salt hydrolase (BSH) deconjugates primary bile acids in the gut [20] and represents one of their resistance mechanisms against bile acid toxicity [48,49]. A recent study by Quraishi et al. proved that mucosa-attaching bacterial genera expressing BSH are more abundant in PSC compared to UC patients without PSC [20]. The presented study detected a significant increase in the BSH producing genera *Roseburia*, *Haemophilus,* and *Fusobacterium* in PSC compared to non-IBD patients, *Roseburia* in PSC compared to UC patients, and *Blautia*, *Ruminococcus,* and *Roseburia* in UC compared to non-IBD patients. An increase of *Roseburia* and *Haemophilus* in PSC patients was also reported in earlier studies [20]. However, the BSH-producing genus *Bacteroides* is significantly lower in PSC compared to UC and non-IBD patients, which is a known finding in PSC, as mentioned above [40]. Therefore, the increase in the BSH producers might be a compensatory mechanism for the decrease in *Bacteroides*.

We have previously demonstrated that the SNP rs1893217 of the PTPN2 gene (which introduces a C-allele in the gene) promotes intestinal inflammation as well as a more severe disease course in IBD [50,51]. Furthermore, in the presence of the PTPN2 variant, distinct alterations of the mucosa-associated gut microbiome were detected in IBD patients, suggesting an interplay of genetic risk factors, intestinal microbiota, and disease course [21]. After the exclusion of patients with a C-allele in the PTPN2 gene, *Brachyspira* was elevated in PSC versus UC patients in our study. Increased mucosal *Brachyspira* colonization has also been detected in patients with irritable bowel syndrome (IBS), especially in IBS with predominant diarrhea [52]. In IBS, *Brachyspira* was associated with the induction of inflammatory mediators in the intestinal mucosa as well as mast cell activation and the induction of the actin related protein (ARP) 2/3 complex, a protein facilitating bacterial adhesion and invasion [52]. The ARP2/3 complex is involved in actin nucleation, a crucial process for cell migration and cytoskeletal motion, which is one of the mechanisms involved in cancer cell migration [53]. Thus, the subunit 4 (ARPC4) is associated with reduced survival in patients with hepatocellular carcinoma [54,55]. In addition, the expression of ARP3 is higher in inflamed intestinal samples of UC patients compared to healthy controls and it is related to epithelial apoptosis in vitro [56]. In contrast, the inhibition of the ARP2/3 complex activated the NF-κB pathway, causing a hypersensitive reaction to osmotic stress, which is associated with IBD [57]. Whether activation of ARP2/3 is protective or disease-aggravating in PSC, and if *Brachyspira* contributes to this process, remains to be elucidated. Analyzing the gut microbiome changes according to sampling site, the genera *Eubacterium* and *Tepidimonas* were significantly higher in the right colon of PSC patients without the C allele vs. control patients. *Eubacterium* are butyrate and BSH-producing bacteria that are associated with beneficial effects on gut and liver [49,58]. This is in line with the assumption that the absence of the PTPN2 risk allele is associated with a less-severe disease course in IBD [50]. Studies with *Tepidimonas* are scarce; however, a recent study detected *Tepidimonas* in extracellular vesicles of pancreatic tumors [59], demonstrating a mechanism by which the gut microbiome might influence extraintestinal organs. Whether this mechanism also applies to PSC, and if it might play a protective, or rather harmful, role in its pathogenesis is another unanswered question.

Despite the detected microbial differences that can be linked to intestinal and/or bile duct and liver inflammation, results need to be interpreted with caution, as the number of PSC patients in this study was rather small. Moreover, the investigated groups slightly differed in smoking status and BMI range. However, as published previously, smoking status did not influence the microbiome in UC patients [26].

## 5. Conclusions

In conclusion, this study underlines the presence of mucosa-associated microbiome changes in the gut of patients with PSC. These changes might be indirectly caused by altered bile acid synthesis or even directly be associated with mucosal inflammation and periductular fibrosis in PSC. These findings may lay the base for precision medicine studies investigating the impact of specific bacterial strains in the pathogenesis of PSC. The presence of a potentially pathogenic gut bacterium in patients without a genetic variation in PTPN2 rather weakens its role as a risk gene for PSC.

## Figures and Tables

**Figure 1 microorganisms-09-01752-f001:**
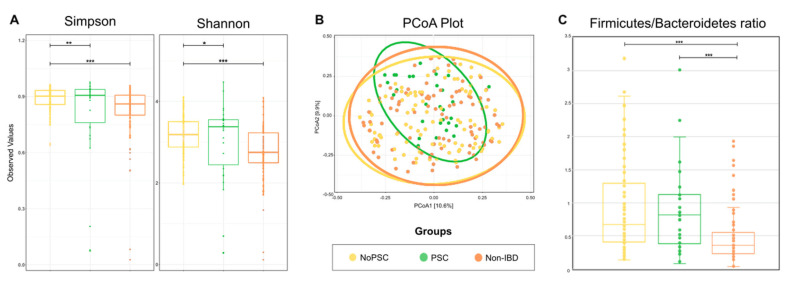
Comparison of the microbial composition between groups. (**A**) α-diversity was compared using the Simpson and Shannon indices in the indicated groups. (**B**) β-diversity was compared by a principal coordinates analysis (PCoA) of the Bray-Curtis dissimilarity between groups. Ellipsoids in PCoA plots represent the 95% confidence interval surrounding each group. (**C**) Firmicutes/Bacteroides ratio in the indicated groups. The boxplots represent the median and the interquartile range. Differences were determined using a one-way ANOVA and marked as follows: * *p* < 0.05; ** *p* < 0.01; *** *p* < 0.001; PSC: primary sclerosing cholangitis; IBD: inflammatory bowel disease. PSC: *n* = 30, non-PSC: *n* = 94, non-IBD: *n* = 92.

**Figure 2 microorganisms-09-01752-f002:**
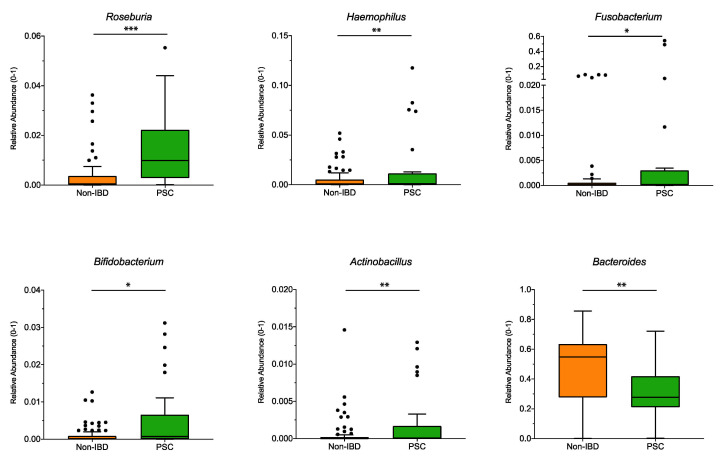
Genus differences between PSC and non-IBD: The relative abundance of significantly different genera (*p* < 0.05; *q* < 0.05) is shown for pooled samples for PSC and non-IBD as indicated. The boxplots represent the median and the interquartile range. The adj-*p* value was reported with *** adj-*p* < 0.001; ** adj-*p* < 0.01; * adj-*p* < 0.05.; PSC: primary sclerosing cholangitis; IBD: inflammatory bowel disease. PSC: *n* = 30, non-IBD: *n* = 92.

**Figure 3 microorganisms-09-01752-f003:**
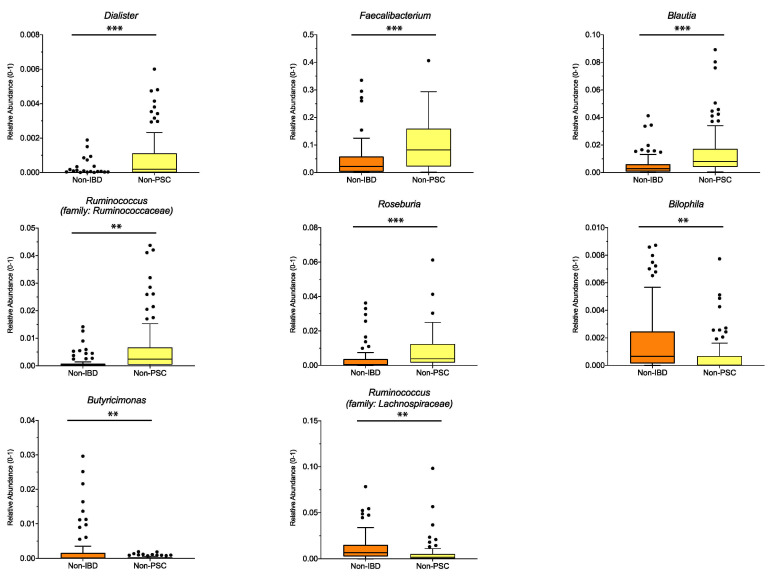
Genus differences between non-PSC and non-IBD: The relative abundance of significantly different genera is shown for pooled samples in non-PSC and non-IBD as indicated. The boxplots represent the median and the interquartile range. The adj-*p* value was reported with *** adj-*p* < 0.001; ** adj-*p* < 0.01; PSC: primary sclerosing cholangitis; IBD: inflammatory bowel disease. Non-PSC: *n* = 94, non-IBD: *n* = 92.

**Figure 4 microorganisms-09-01752-f004:**
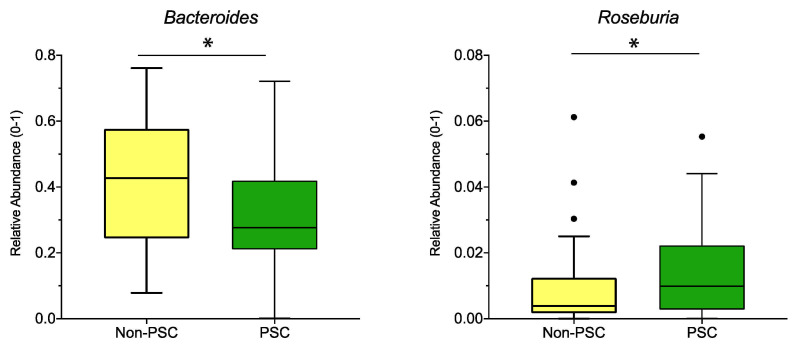
Genus differences between PSC and non-PSC: The relative abundance of significantly different genera (*p* < 0.05; *q* < 0.05) is shown for PSC, non-PSC, and non-IBD as indicated. The boxplots represent the median and the interquartile range. The adj-*p* value was reported with * adj-*p* < 0.05.; PSC: primary sclerosing cholangitis; IBD: inflammatory bowel disease. PSC: *n* = 30, non-PSC: *n* = 94.

**Table 1 microorganisms-09-01752-t001:** Demographics of study subjects. BMI: body mass index; PSC: primary sclerosing cholangitis; IBD: inflammatory bowel disease; MTWAI: Modified Truelove and Witts activity index; PTPN2: protein tyrosine phosphatase nonreceptor type 2.

	PSC (*N* = 7)	Non-PSC (*N* = 42)	Non-IBD (*N* = 28)
BMI (kg/m^2^); median (range)	23.8, (21.3–48.8)	23.8, (17.6–35.9)	24.2, (17.3–29.0)
Age at enrollment (year); median (range)	32, (19–49)	39, (21–58)	41, (20–49)
Smoker at enrollment, *N* (%)	0 (0%)	6 (14%)	5 (19%)
MTWAI; median, (range)	3, (0–10)	2, (0–9)	-
PTPN2 variant:			-
TT (*N*, %)	5, 71%	24, 57%	
CT (*N*, %)	0, 0%	6, 14%	
CC (*N*, %)	0, 0%	1, 2%	
Unknown (*N*, %)	2, 29%	11, 26%	

## Data Availability

16S sequencing data of the SIBDCS can be found at https://doi.org/10.6084/m9.figshare.7335068.v1 (accessed on 19 December 2018). Metadata of PSC patients reported in this study was uploaded to https://figshare.com/articles/dataset/PSC/14923059 (accessed on 7 July 2021).

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
