# Peer review of "Modulation of the Mucosa-Associated Microbiome Linked to the PTPN2 Risk Gene in Patients with Primary Sclerosing Cholangitis and Ulcerative Colitis"

_microorganisms, 2021, doi:10.3390/microorganisms9081752_

Round 1

Reviewer 1 Report

line 38 - too generally - this is not "function changes" applies to the gut-liver axis. Too general sentence supported by one reference. If you make a reference to the axis, mention the mechanism of its action
line 40 - bacterial metabolites cannot be considered as harmful substances that end up in the liver. Products of the basic metabolism of bacteria (it is assumed that about 30% circulate in the peripheral blood) - what determines the consequences is whether they are pathogenic or beneficial bacteria
line 59- there is no clear research hypothesis
line 67 - why are PSC patients not a representative group compared to the rest?
line 92 - was DNA analysis performed only with nanodrop? why before sequencing was not performed an analysis on a bioanalyzer, or even agarose, can we confirm the lenght of the isolated DNA or the integrity of the band?
line 95 - why such a discrepancy in concentration? why the trials were not even?
Table 1- based on the differences in BMI of the analyzed patients, the composition of the microbiota may be influenced by many other factors, not only the disease itself. The group of 7 patients seems to be too small with this BMI spread, especially for the PSC group. Additionally, the age difference between the PSC and the non-IBD group is quite large. Smoking is also important for the microbiota. There is a lack of robust data on patients that can exclude or allow to refute speculation that the data shown is actually related to the disease entity.

Author Response

Please find below our responses to the reviewers’ comments marked with blue letters.

Reviewer 1:

Comments and Suggestions for Authors

line 38 - too generally - this is not "function changes" applies to the gut-liver axis. Too general sentence supported by one reference. If you make a reference to the axis, mention the mechanism of its action

We have adjusted the sentence and explained the term “gut-liver axis” in more detail.

line 40 - bacterial metabolites cannot be considered as harmful substances that end up in the liver. Products of the basic metabolism of bacteria (it is assumed that about 30% circulate in the peripheral blood) - what determines the consequences is whether they are pathogenic or beneficial bacteria

«Microbial products» not only means meatbolites, but also pathogen associated molecular pattern, which basically, besides metabolites, also includes bacterial components, for example LPS. We have adjusted this sentence.

line 59- there is no clear research hypothesis

We have adjusted the hypothesis and aims.

line 67 - why are PSC patients not a representative group compared to the rest?

The Swiss IBD cohort study primarily assembles data from patients with inflammatory bowel disease (IBD). Though we included more than 3000 patients in the cohort, biopsies for further analyses were only biobanked in about 400 patients. PSC is a rare liver disease, so biopsies for 16S sequencing were only available from 7 patients with PSC for microbiome analyses. We have added this information to the paper.

line 92 - was DNA analysis performed only with nanodrop? why before sequencing was not performed an analysis on a bioanalyzer, or even agarose, can we confirm the lenght of the isolated DNA or the integrity of the band?

DNA concentration and purity were tested with NanoDrop. DNA extraction was carried out according to the manufacturer’s protocol (AllPrep DNA/RNA extraction kit from qiagen). As the reviewer suggested, we have tested some of the samples’ integrity before starting the study analyses. Hence, we know that samples are in good quality and they amplified in PCR for V5-V6 region of 16S rRNA.

line 95 - why such a discrepancy in concentration? why the trials were not even?

In this study we have used biopsies (and not stool samples) to extract 16S rRNA. Biopsies generally contain 95-99.5% host DNA, i.e., a very small amount of this DNA is microbial DNA – according to our shotgut metagenomic studies (Yilmaz, B., et al., Microbial network disturbances in relapsing refractory Crohn's disease. Nat Med, 2019. 25(2): p. 323-336). Therefore, some samples required higher amount of DNA loading for PCR. We therefore amplified DNA with different concentrations. However, we state that “26 pM concentration was used for sequencing” to normalize the data for statistical analyses.

Table 1- based on the differences in BMI of the analyzed patients, the composition of the microbiota may be influenced by many other factors, not only the disease itself. The group of 7 patients seems to be too small with this BMI spread, especially for the PSC group. Additionally, the age difference between the PSC and the non-IBD group is quite large. Smoking is also important for the microbiota. There is a lack of robust data on patients that can exclude or allow to refute speculation that the data shown is actually related to the disease entity.

As pointed out above, 7 was the maximum of PSC patients with available data from the cohort and we did our best to match the Non-PSC and Non-IBD groups to the PSC patients. Importantly, these patients were extensively characterized in Yilmaz, B., et al., Microbial network disturbances in relapsing refractory Crohn's disease. Nat Med, 2019. 25(2): p. 323-336, for most of the clinical parameters, such as BMI and smoking, inflammation status and disease treatment. We have included this knowledge into the matching of patients for microbial analysis. For instance, in patients with UC, we did not detect an influence on the microbiome by smoking status. However, we understand the reviewer’s concerns and have added a comment on these differences as a limitation of the study in the Discussion section.

Reviewer 2 Report

The present study is interesting and of interest to the medical and scientific community.

After reading it I have some comments and questions to ask.

Minor:

Avoid abbreviations in the title

After a period, do not start a sentence with a number in digits, but with a letter.

The abstract is unintelligible due to multiple undefined abbreviations.

Write s instead of sec for seconds

Many abbreviations lack definition

Sometimes Alpha is used and sometimes α. Unify the criteria

Include abbreviations in tables and figures

Specify the value of n in the figures and whether the error or standard deviation is represented when applicable.

Mayor:

I do not understand the statement "3.5. Influence of the PTPN2 SNP rs1893217".

Where are these results represented?

I think the part of UC patients carrying the C-allele 195 of the PTPN2 SNP rs1893217 is missing or should be supported with more studies and determination of correlations.

 A PERMANOVA analysis to test differences between OTUs abundances should be done.

Has medication and its possible influence on the microbiota profile been taken into account?

Did the patients present other comorbidities?

Author Response

Reviewer 2:

Comments and Suggestions for Authors

The present study is interesting and of interest to the medical and scientific community. After reading it I have some comments and questions to ask.

Minor:

Avoid abbreviations in the title

We have replaced most of the abbreviations in the title with unabbreviated name.

After a period, do not start a sentence with a number in digits, but with a letter.

We have adjusted this sentence in the abstract.

The abstract is unintelligible due to multiple undefined abbreviations.

We have introduced the abbreviations and hope that this will be possible to keep, although this leads to an exceedance of words in the abstract.

Write s instead of sec for seconds

We have replaced ”sec” with “s” in the methods section.

Many abbreviations lack definition

We have added definitions for the abbreviation “IBD” (inflammatory bowel disease), “BMI” (body mass index), “OTU” (operational taxonomic unit) and erased unnecessary abbreviations.

Sometimes Alpha is used and sometimes α. Unify the criteria

We have unified the terms and use “α” throughout the manuscript.

Include abbreviations in tables and figures

We have added the definitions for the abbreviations in table 1 and the figure legends.

Specify the value of n in the figures and whether the error or standard deviation is represented when applicable.

We have added this information in the figure legends.

Mayor:

I do not understand the statement "3.5. Influence of the PTPN2 SNP rs1893217". Where are these results represented?

Since we had only a few patients carrying the C-allele, we could not detect meaningful differences between these patients and the patients without the allele. So we excluded these patients and analyzed group differences without the patients carrying the (potentially disease aggravating) C-allele. Results are presented within the text. As there was just a difference in the abundance of Brachyspira, we decided to not add another figure to this result. To not confuse the reader with  the subheading, we exchanged the title to “Results according to genetic variation ofof the PTPN2”.

I think the part of UC patients carrying the C-allele 195 of the PTPN2 SNP rs1893217 is missing or should be supported with more studies and determination of correlations.

As stated above, we did not have enough patients carrying the C-allele to do analyses with these patients.

A PERMANOVA analysis to test differences between OTUs abundances should be done.

We thank to reviewer for this suggestion. We have used MaAsLin2 pipeline by Huttenhower’s group which is a well-documented and extremely comprehensive pipeline. To the best of our knowledge, it is well-accepeted in the computational microbiota analysis community, as it efficiently determines multivariable associations between phenotypes, environments, exposures, covariates and microbial meta’omic features. This pipeline uses general linear models to accommodate most modern epidemiological study designs, including cross-sectional and longitudinal, and offers a variety of data exploration, normalization, and transformation methods.

Has medication and its possible influence on the microbiota profile been taken into account?

We have included the medication for treatment of UC into the analysis but we didn’t see any major contribution on the profile and hence we didn’t report this in details.

Did the patients present other comorbidities?

Unfortunately, we do not have further information on that.

Round 2

Reviewer 2 Report

Although very briefly, the authors have answered most of my questions.

On the other hand, if the abstract exceeds the number of words, it is up to the authors to summarize to the appropriate length.